# Assessment of Soil Salinity Changes under the Climate Change in the Khorezm Region, Uzbekistan

**DOI:** 10.3390/ijerph19148794

**Published:** 2022-07-20

**Authors:** Mukhamadkhan Khamidov, Javlonbek Ishchanov, Ahmad Hamidov, Cenk Donmez, Kakhramon Djumaboev

**Affiliations:** 1Department of Irrigation and Melioration, “Tashkent Institute of Irrigation and Agricultural Mechanization Engineers” National Research University (“TIIAME” NRU), Kary-Niyaziy 39, Tashkent 100000, Uzbekistan; khamidov_m@mail.ru (M.K.); ahmad.hamidov@zalf.de (A.H.); 2Research Area 3 “Agricultural Landscape Systems”, Leibniz Centre for Agricultural Landscape Research (ZALF), Eberswalder Straße 84, 15374 Müncheberg, Germany; cenk.doenmez@zalf.de or; 3Landscape Architecture Department, Remote Sensing and GIS Lab, Cukurova University, Adana 01330, Turkey; 4Regional Representative Office for Central Asia, International Water Management Institute (IWMI), Apartment 120, House 6, Osiyo Street, Tashkent 100000, Uzbekistan; k.djumaboev@cgiar.org

**Keywords:** soil, salinization, projection, homogeneity test, water-energy-food nexus, Uzbekistan

## Abstract

Soil salinity negatively affects plant growth and leads to soil degradation. Saline lands result in low agricultural productivity, affecting the well-being of farmers and the economic situation in the region. The prediction of soil salinization dynamics plays a crucial role in sustainable development of agricultural regions, in preserving the ecosystems, and in improving irrigation management practices. Accurate information through monitoring and evaluating the changes in soil salinity is essential for the development of strategies for agriculture productivity and efficient soil management. As part of an ex-ante analysis, we presented a comprehensive statistical framework for predicting soil salinity dynamics using the Homogeneity test and linear regression model. The framework was operationalized in the context of the Khorezm region of Uzbekistan, which suffers from high levels of soil salinity. The soil salinity trends and levels were projected under the impact of climate change from 2021 to 2050 and 2051 to 2100. The results show that the slightly saline soils would generally decrease (from 55.4% in 2050 to 52.4% by 2100 based on the homogeneity test; from 55.9% in 2050 to 54.5% by 2100 according to the linear regression model), but moderately saline soils would increase (from 31.2% in 2050 to 32.5% by 2100 based on the homogeneity test; from 31.2% in 2050 to 32.4% by 2100 according to the linear regression model). Moreover, highly saline soils would increase (from 13.4% in 2050 to 15.1% by 2100 based on the homogeneity test; from 12.9% in 2050 to 13.1% by 2100 according to the linear regression model). The results of this study provide an understanding that soil salinity depends on climate change and help the government to better plan future management strategies for the region.

## 1. Introduction

Climate change is becoming a crucial factor that can significantly impact all areas of human activity. It has a negative impact on the environment in many parts of the planet, and on the lives and health of the population in various sectors of the economy [1]. The impact of climate change on agriculture, in particular, is high, as agriculture is one of the most weather-dependent sectors of the economy. The negative impact of climate change associated with rapid melting glaciers and extended drought periods has resulted in a significant decline in agricultural development, especially in the semi-arid regions of the world [2,3].

In Uzbekistan—the focus of this research—climate change has resulted in alterations in air temperature and rainfall. This, in turn, has led to agriculture being the most vulnerable sector due to the lack of effective adaptation measures [4]. There are two main rivers in Uzbekistan—Amudarya and Syrdarya—that feed about 80% of irrigated lands. There are projections that Amudarya’s water flow will be reduced by 15% and Syrdarya’s by 5% by 2050 [5]. This reduction in water flow will significantly affect agricultural productivity and soil quality.

One of the important consequences of water flow reduction is soil salinity [6]. It is a significant problem that mainly affects environmental quality, agricultural management, and productivity [7]. The accumulation of salts in the root zones may reduce the development of crops, deteriorate the irrigation water quality, and decrease crop yields [8]. The main impact of soil salinization is the deterioration of various ecological services, which negatively affects the soil structure, nutrient cycle, and crop productivity [9,10].

Analysis of the salinity levels of irrigated lands is particularly important in obtaining high crop yields. Soil salinity has affected over 50% of irrigated lands in Uzbekistan [11]. In fact, there are two major causes of salinity in the world: natural (primary salinization) and human-made (secondary salinization) [12]. Mismanagement of water and land resources over the last 40 years has led to increased soil salinity in Uzbekistan [13]. Moreover, secondary salinization is aggravated under poor drainage settings [14]. Some regions have been using collector-drainage water that has poor quality to irrigate agricultural fields in response to emerging climate change and water scarcity [15]. Therefore, one of the important issues is to analyze and predict the impact of climate change on land reclamation.

Accurate projection and continuous monitoring of soil salinity and land reclamation are essential for preventing their potential negative effects on ecosystem productivity and restoration. Monitoring of soil salinization can provide detailed information on its fluctuations [16]. However, the collected information should be evaluated carefully, representing the distribution of soil salinity and its potential changes under climate change across scales. Recently, the evaluation of meteorological changes and sophisticated techniques (i.e., remote sensing and multi-variate statistical methods) have commonly been applied for monitoring and assessing soil salinity at the regional scale. In 2012, Eldeiry and Garcia applied various statistical analyses, including linear and non-linear kriging techniques, to predict soil salinity with spatial data [17]. Soil salinity trends have been analyzed using linear models, semivariograms, and interpolation techniques as valuable tools to provide relevant information on soil quality and its management across scales [17,18,19]. Zhang et al. [20] found that multi-variate analysis in a semi-humid irrigated district was efficient in examining the relationship between soil salinity and environmental factors (i.e., topography, water table). Several studies have analyzed the distribution of saline soils using measurement data and spatial interpolation [21,22,23,24], mainly spatial at a global scale [25]. Studies have often overlooked the effects of climate change on salinity dynamics in complex regions; thus, the focus of recent studies showed that there is still a need for combined analysis evaluating long-term variations in the soil salinity at a local scale, especially in data-scarce semi-arid regions.

As a result of transpiration, the plant roots release fresh water, leaving salts. Similarly, evaporation from the soil surface releases soil water into the atmosphere, leaving salts. Then, when the water evaporates, the salts in the water accumulate, which leads to the accumulation of salts on the soil surface. Climate change, that is, an increase in air temperature, leads to an increase in water evaporation and an increase in soil salinity [1]. Addressing these climatic changes and assessing the effects of air temperature on soil salinity is essential for efficient decision-making for sustainable agricultural management.

Addressing this need, this study aimed to analyze and predict the soil salinity dynamics and their long-term trends under the influence of climate change in the Khorezm region in Uzbekistan. The long-term changes in the soil salinity from 1991 to 2020 were assessed using the homogeneity test and linear regression model to examine the relationship between the climatic variables and salinity levels of the soil at the study site.

Analysis of soil salinity using statistical methods, particularly linear regression, has created great potential among other methods to improve soil salinity modeling due to its fastness, practicality, and cost-effectiveness [26]. Regression is a statistical tool that is widely used by scholars to describe the nature of the relationship between variables. Consequently, these relationships can be positive or negative, linear or non-linear [27]. In regression, variables are divided into independent variables and dependent variables. A dependent variable is a response variable that can be predicted by the independent variable. Regression models are developed to predict certain variables based on other predictor variables [28]. They are developed using many techniques such as simple linear regression, multiple linear regression, non-linear regression, non-parametric regression, and multivariate regression. For instance, Clark used multivariate nonparametric regression to predict the future state of highway traffic [29]. Goyal and Goyal applied artificial neural engineering and regression to predict the shelf life of an instant coffee drink [30]. Moreover, the homogeneity test was chosen to detect homogeneity in the time series analyzed in this study [31]. A non-parametric approach to change point analysis was proposed, which is still widely used. This test detects mean changes and calculates their significance in hypothesis testing [32].

This study’s hypothesis was that as the air temperature increases, the medium and strong salinity areas of the lands would increase, whereas slightly saline areas would decrease [33]. Given the importance of analyzing soil salinity fluctuations, this study can provide valuable outputs to decision-makers for sustainable soil and land use in arid regions. It shows excellent potential for multi-variate statistical approaches by integrating climatic factors and soil salinity information for long-term soil and agricultural research.

## 2. Study Area and Data

### 2.1. Case Study Area

The Khorezm region, selected for the research, is located in the northwest of Uzbekistan, in the lower reaches of the Amudarya, 60–62° east longitude, 40–42° north latitude (Figure 1). It is bordered by the Republic of Karakalpakstan to the north and the Bukhara region to the south. The land area is about 6050 km^2^ and occupies roughly 1.34% of the territory of Uzbekistan. As of January 2022, the population of the Khorezm region is about 1.9 million people, an increase of 26,800 people or 1.43% compared to January 2021 [34].

The territory stretches 280 km from northwest to southeast and 80 km from west to east in the latitude of Urgench. The most northern point of the region is the Nuronbobo grove near the village of Olchin, Gurlen district. The most southern point is located a little south of Tupraqqala.

The region is located in the northern part of the Turan lowland, occupying part of the left bank of the ancient Amudarya delta and a small part of the Kyzylkum desert on the right bank. It is located in the lower reaches of the Amudarya River and is a large alluvial plain in terms of relief and has an average slope of 0.0003–0.0005. The Khorezm region is divided into two parts in terms of land structure: the large northern part, which is about 100–110 m above sea level, and the southern part, which is 120–150 m above sea level. The entire territory of the region is mainly occupied by plains and small hills.

The region suffers from high levels of soil salinity, water shortage, energy inefficiency for pumping irrigation and low level of food production, which is a typical water-energy-food-environment (WEFE) nexus challenge. Since the Khorezm region’s irrigated lands are 100% saline, the samples were taken from three different districts with three different salinity levels: slightly saline areas in Khiva district, moderate saline areas in Kushkupir district, and high saline areas in Shavat district. Moreover, soil samples were taken from 20 points with 5 different depths (0–10, 10–20, 20–30, 30–60, 60–90 cm) from each area. Soil analyses were carried out in the laboratory of the Research Institute of Soil Science and Agrochemistry. The average values of the physical and chemical properties of the soils of the Khorezm region are given in Table 1.

The physical and chemical properties of the soils are important factors that directly affect soil salinity. Salinity can also affect these properties by saline soil water, causing fine particles to bind together into aggregates. This is known as flocculation, which positively affects soil aggregation and stabilization through aeration and root growth; however, high salinity levels can have adverse and potentially lethal effects on plants [35]. In this process, soil texture plays a vital role in salinity, mainly through irrigation. It determines the amount of water stored or passing and the ability of salt to bind the soil. For instance, clay soils are slower to drain than coarse-textured soils due to their smaller particles blocking the spaces and preventing water percolation. In usual irrigation practices, sandy soils can flush more water through the root zone by larger particles than clay soils. Thus, sandy soils can handle irrigation with high-saline water by removing dissolved salt from the root zone by leaching. A lower proportion of the clay texture than sandy soils can be advantageous for potential salt leaching in the study region.

Another factor of soil salinity is soil pH. Generally, an increasing salinity level is not expected to influence pH. The pH of saline soils is mainly below 8.5 because of the prevention of dispersion of soil colloids through salt [36]. In this context, the pH values of the Khorezm region are primarily reasonable in different depth levels where they are lower than the salinity indication threshold.

The climate of the region is sharply continental, meaning that summers are hot and dry, and winters are cold. In winter, the average temperature is −6 °C lower than in the rest of the south and east of Uzbekistan. The average annual temperature is +12 °C; the average temperature in January is −5 °C and the average temperature in July is +30 °C. The absolute temperature is a minimum of −32 °C and the absolute temperature is a maximum of +45 °C. The average annual rainfall in the region is 78–79 mm (the main part of precipitation falls in spring and autumn), and the growing season lasts 200–210 days. Climatic conditions are of great importance in the cultivation of crops. Since the territory of the Khorezm region is covered with sand, the temperature rises to +43 and +45 °C in summer. The lowest temperature in winter is −30 or −33 °C. Hot summers, cold winters, sudden changes in the weather during the day, low rainfall, and dry weather are the main features of the regional climate. Due to its complex climate and landscape, the Khorezm region is an ideal testbed for monitoring and evaluating soil salinity level detection using a statistical framework. Soil salinity levels were analyzed by classifying slightly, moderately, and highly saline soils.

### 2.2. Data

The data on the temperature regime of the region was derived from the meteorological weather station Khiva for the period from 1928 to 2020 and the weather station Urgench for the period from 1930 to 2020. It comprised long-term annual and monthly averages to determine general temperature trends at the study site. These data were provided by the Center of Hydrometeorological Service of Uzbekistan through personal communication. Perennial data were transferred from handwritten form to electronic form and analyzed.

The observation data was obtained from the Khiva (for the period of 1928–2020) and Urgench meteorological stations (1930–2020). The geographic coordinates of Khiva are latitude: 41.38° N and longitude: 60.36° E, with an elevation above sea level of 86 m. The latitude and longitude of Urgench are 41.58° N and 60.63° E and its elevation is 101 m.

Data on the soil salinity level for 1991–2020 was obtained from the Khorezm Ameliorative Expedition. Salinity areas were analyzed over 5-year periods (Table 2).

## 3. Methods

The methodology for projecting and evaluating the trends in salinity changes in the Khorezm region comprised a comprehensive multivariate statistical analysis, including the homogeneity test.

### Homogeneity Test

Checking homogeneity in climate research is crucial to represent fundamental changes in weather and climate. Inhomogeneity in climate data can occur for several reasons, including instrument error, changes in adjacent areas of the instrument, and human misuse. If the homogeneity is not tested before analyzing the trend, the results indicate erroneous trends.

In this study, the absolute homogeneity tests were performed on individual station records. Relative homogeneity tests were performed by generating a reference series using the XLSTAT software package and calculating the ratio of the observed series to the reference series [37,38,39]. We tested the linear regression and homogeneity test using XLSTAT software. For this software, we used the mean annual air temperature and soil salinity levels. Here, a common statistical test (Pettitt’s test) was used [31]. The Pettitt’s test is a nonparametric test that requires no assumption about the data distribution. The Pettitt’s test is an adaptation of the tank-based Mann–Whitney test, which allows identification of the time at which the shift occurs.

In his article of 1979, Pettitt describes the null hypothesis as being that the *T* variables follow the same distribution *F*, and the alternative hypothesis as being that at a time *τ* there is a change in the distribution. Nevertheless, the Pettitt test does not detect a change in the distribution if there is no change in the location. For example, if, before the time *τ*, the variables follow a normal *N* (0,1) distribution and from time *τ*, an *N* (0,3) distribution, the Pettitt test will not detect a change in the same way a Mann–Whitney would not detect a change in the position in such a case. In this case, one should use a Kolmogorov–Smirnov-based test or another method to detect a change in a characteristic other than the location.

Addressing the aim of this study, the homogeneity of the data over the observation period and the trend components in the mean air temperature data were examined. This allowed the moment t_c_, if any, to be determined by noting the shift in the mean annual air temperature, assuming a statistically significant difference between the mean temperature for the period before and after the point of change in t_c_.

Yearly test results for homogeneity in the temperature data series show that points of change were found in 1928 and 1984. The test for homogeneity shows a change point of 1984, since this was the year that change occurred at the weather station in Khiva. The homogeneity model is a non-parametric test, which means that its application does not require assumptions about the distribution of the data. This test estimates the null hypothesis H_0_, which assumes that the data are homogeneous throughout the observation period, i.e., that the data come from one or more distributions with the same location parameter (means). The alternative hypothesis H_1_ implies the presence of a non-random component among the data causing the location parameter to shift at a particular moment. In addition to providing a test of data homogeneity, the homogeneity test also determines—if the alternative hypothesis is considered acceptable—the point of change when the location parameter shift occurred.

An appealing non-parametric test to detect a change is to use a version of the Mann–Whitney two-sample test. Let:(1)$$D_{ij}=sgn\left(X_{i}-X_{j}\right)$$
where *sgn*(*x*) = 1 if *x* > 0, 0 if *x* = 0, −1 if *x* < 0, then consider:(2)$$U_{t,T}=\sum_{i=1}^{t}\sum_{j=i+1}^{T}D_{ij\;}\;$$

The statistic $U_{t,T}$ is equivalent to a Mann–Whitney statistic for testing that the two samples *X_i_*,…, *X_t_* and *X_t+_*_1_,…, *X_T_* come from the same air temperature. The statistic *U_t_*_,*T*_ is then considered for the value soft with 1 < *t* < *T*. We propose for the test of H: no change against A: change, the use of the statistic:(3)$$K_{T}=\underset{1<t<T}{\max}\left|U_{t,T}\right|\;$$

As a result of this function, we obtain that the future increasing or decreasing quantities form mathematically based values:(4)$$y\left(x\right)=y_{k-1}+\frac{x_{0}-x_{k-1}}{x_{k}-x_{k-1}}\left(y_{k}-y_{k-1}\right)$$

Prognostic extrapolation is based on the mean approach of mean forecasting. All future values of the time series studied with this approach are equal to the mean value:(5)$$\bar{y}=\frac{y_{1}+y_{2}+y_{3}\ldots y_{T}}{T}$$
where $\left(y_{1}+y_{2}+y_{3}\ldots y_{T}\right)$ are the regional-scale values of the observed soil salinity, *T* is the prospective years, and the five-year intervals are relative.

## 4. Results

A comprehensive analysis of the climatic trends in the study sites was assessed in the first step. In this context, we analyzed the mean annual air temperature at the Khiva and Urgench meteorological weather stations based on the homogeneity test. As a result of the average air temperature at the Khiva meteorological station from 1928 to 2020 and at the Urgench meteorological station from 1930 to 2020, the point of temperature change was determined as 1984 at Khiva and 1976 at Urgench. The mean annual air temperature at Khiva was 12.430 and 13.932 °C in the first 1928–1984 period and the second period from 1985 to 2020, respectively. The average air temperatures of the Urgench meteorological station from 1928 to 1976 and the second period from 1977 to 2020 were 12.430 and 13.932 °C, respectively (Figure 2 and Figure 3).

The change in air temperature is mainly affected by global warming. Earth’s temperature has increased by 0.08 °C per decade since 1880, but the rate of warming since 1981 has been more than twice that: 0.18 °C per decade [40]. Since 1970, Steele et al. also detected increasing atmospheric abundances of two other major greenhouse gases: methane (CH_4_) and nitrous oxide (N_2_O) [41]. Changes in air temperature in these areas were determined using expert interviews. The rationale behind the sudden change in air temperature at these meteorological stations is the decrease in the volume of water in the Aral Sea since the 1970–1980s. Due to the lack of water capacity in the air, the temperature has also increased.

Climate change and rising temperatures may increase evapotranspiration, including the evaporation of water from soils. As a result, water evaporates and the salt remains in the soil, increasing the soil salinity. This leads to study of the relationship between climate change and soil salinity (Figure 4). In order to prevent or reduce soil salinity, the Republic of Uzbekistan has adopted state programs to improve the reclamation of irrigated lands, and it is desirable to carry out further improvement.

Assessing the impact of climate change on land reclamation, particularly the salinity of irrigated land, requires complex decisions. For instance, temperature changes cause prolonged heat stress, which leads to decreased groundwater resources and, therefore, increases the saline proportion of inland soils [42]. Thus, the predictions of land salinity allow for better planning of management strategies in the field of land reclamation in these regions.

Following the comprehensive climatic analysis, we predicted that the degree of salinization of irrigated lands would change in the Khorezm region. Based on the linear analysis and homogeneity test, the projection of saline areas at different salinity levels is illustrated in Figure 5. From this figure, we can see that salinity levels increased from 1996 to 2000 due to a lack of funding to improve drainage systems, which may have exacerbated the problem of soil salinity. Starting in 2007, the government took the first step of improving the drainage systems through the Cabinet of Ministers’ Decree No. 3932 and established the Republican Irrigated Land Amelioration Fund. The ultimate goal of the Fund was to allocate funding to maintain and rebuild drainage systems in Uzbekistan. This might have helped to reduce highly saline areas over time. However, with the emerging climate change in the Khorezm region, the salinity areas (e.g., moderately and highly) might increase in the future [43], meaning that potential land abandonment may take place. In fact, White et al. reported that due to soil salinization and water shortages, the overall irrigated area in the Amudarya basin has recently decreased from 4.0 to 3.4 million ha [44].

Both the linear analysis and homogeneity tests showed a significant correlation in slight- and high-saline areas. The moderate-saline areas were projected to show a slight deviation from 2016 to 2050 only. According to the results of the analysis, in 2011–2015, 56% of the irrigated lands of the region were slightly saline soils, 31.6% medium saline, and 12.4% highly saline. Using the methods of statistical analysis, according to the results of the forecast, the situation can be developed in the following way: in particular, according to the linear analysis, slightly saline arable lands will decrease, and highly saline irrigated lands will increase, but moderately saline irrigated lands will stay relatively stable. Increasing salt levels in saline irrigated lands negatively affect the biochemical attributes of crops, which leads to a decrease in biomass and agricultural yield [45]. It usually occurs due to water movement into crop roots by an osmosis process, which is controlled by the salt level in the soil water and the water contained in the plant. In this process, the high salt concentration in the water causes the formation of chlorophyllase, which disturbs photosynthesis, resulting in a limited crop yield by hampering nitrogen absorption [46,47]. In the study region, if in the period up to 2020 slightly saline lands were 57.8%, reduced by up to 55.9% during 2021–2050, and will decrease by up to 54.5% of the area by 2100, then highly saline lands will be 11.9%, 12.9%, and 13.1%, respectively. Moderately saline soils will be 30.3%, 31.2%, and 32.4%, respectively. Based on the estimated salt levels, slight increases in salinity in highly saline lands may accelerate, decreasing the growth rate and resulting in smaller leaves, shorter stature, and sometimes fewer leaves. This will lead to crop yield losses in the study region by interfering in nitrogen uptake, reducing growth, and stopping plant reproduction. As a positive indicator for agricultural production, the linear analysis did not show much variability in the salt levels in the study regions. The results of the homogeneity test also showed the same change fortunately. In particular, in 2021–2050, slightly saline soils will amount to 55.4% and during 2051–2100, about 52.4%, whereas highly saline irrigated areas will, respectively, be 13.4% and 15.1%, and medium saline soils will be 31.2% and 32.5%, respectively. Different salt levels may affect crops differently. For instance, if the concentration of the ions in the soil increases through a salinity increase, the crop productivity may be limited because of poisoning and possible plant death [48]. Moreover, it may lower the seedling growth rate, resulting in a decrease in crop productivity because of a decreased soil water potential [49,50]. Therefore, the limited salt variability in the results can be interpreted as a potential positive factor in agricultural production.

Moreover, the results of the homogeneity test applied in our study indicated a linear relationship between the climatic factors and the soil salinity locally as it is evident in semi-arid and arid regions of the world, where changes in precipitation and temperature significantly influence soil salinity [51]. Moreover, irrigation with saline water, low precipitation, and high evapotranspiration are vital factors that cause salinization at 10% annually in agricultural lands [52]. The results of the soil salinity levels and outcomes align with many studies in similar regions [24,53].

Going forward, it would be interesting to investigate how the recent trend regarding technology implementation in the region—namely, water-saving irrigation technologies—has affected soil salinity. These technologies are heavily promoted and subsidized by the government in response to potential climate change and water scarcity in Uzbekistan [54].

## 5. Conclusions

The implementation of multi-variate statistical approaches to estimate and evaluate soil salinity is useful for better decision-making in the management of agricultural regions. In our study, the dynamics of the salinity of irrigated lands in the Khorezm region under the influence of climate change were assessed and predicted based on linear and homogeneity tests. The results indicate that slightly saline areas would generally be reduced in compensation for an increase in moderately and highly saline areas. This shows an evident expansion of salinity levels across the region. Based on these projections, Uzbek policymakers should develop effective measures for changes in salinity levels, exacerbated by the emerging climate change in the region. The results presented here should also encourage local actors to monitor soil parameters regularly and use resources (e.g., water and fertilizers) more sustainably. Finally, the impact of anthropogenic factors (population growth, irrigation, fertilization, and environmental flows) as well as WEFE nexus challenges should be considered in future studies.

## Figures and Tables

**Figure 1 ijerph-19-08794-f001:**
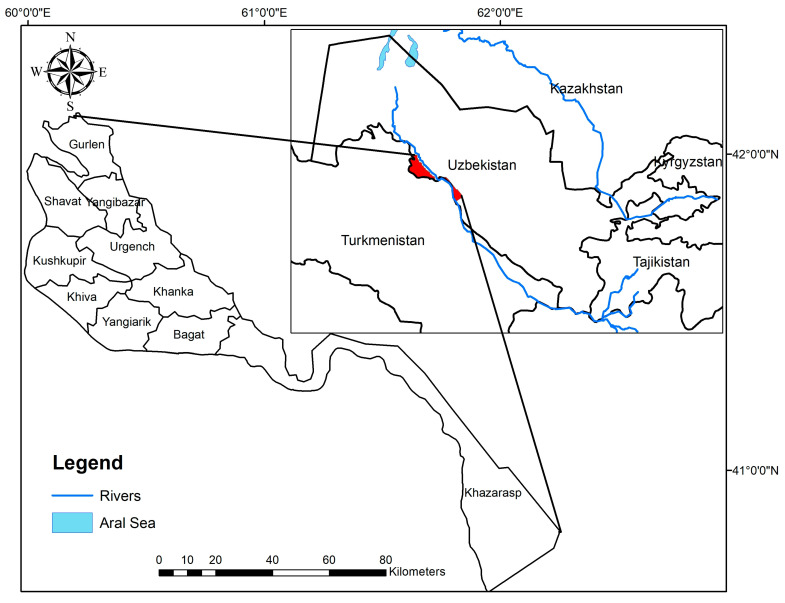
Location of the study region (own graph, generated using ArcGIS).

**Figure 2 ijerph-19-08794-f002:**
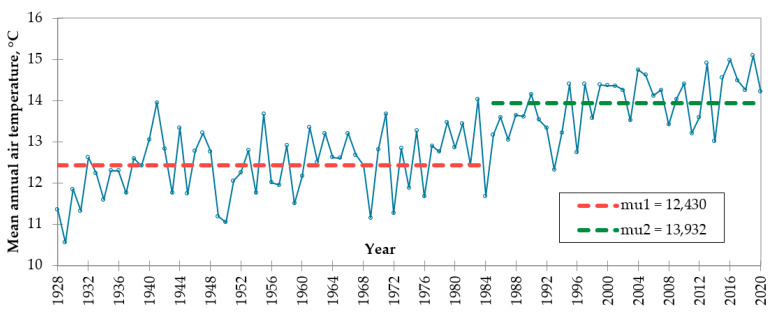
Mean annual air temperature at the Khiva meteorological station in 1928–2020 (point of change in 1984).

**Figure 3 ijerph-19-08794-f003:**
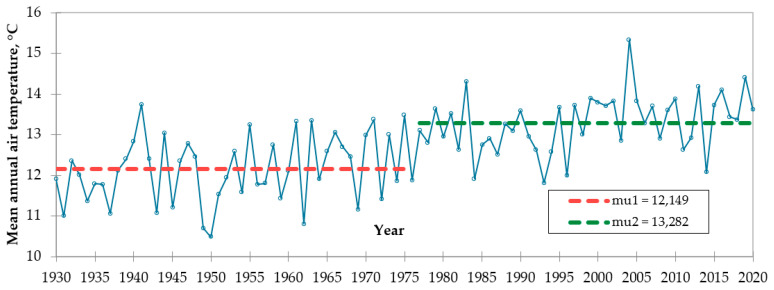
Mean annual air temperature at the Urgench meteorological station in 1930–2020 (point of change in 1976).

**Figure 4 ijerph-19-08794-f004:**
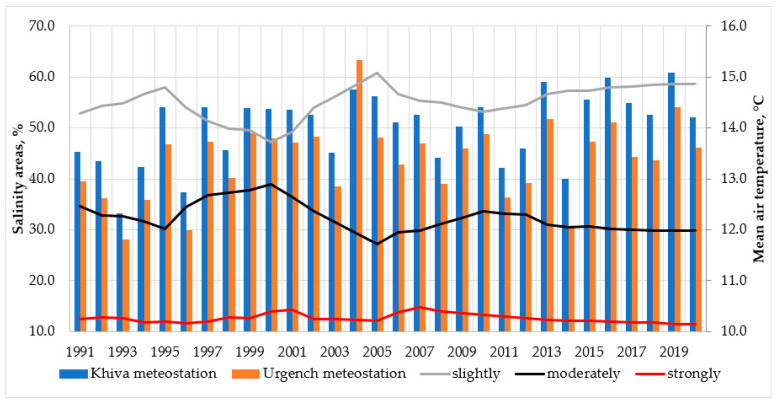
Dynamics of soil salinity and average air temperature change from 1991 to 2020.

**Figure 5 ijerph-19-08794-f005:**
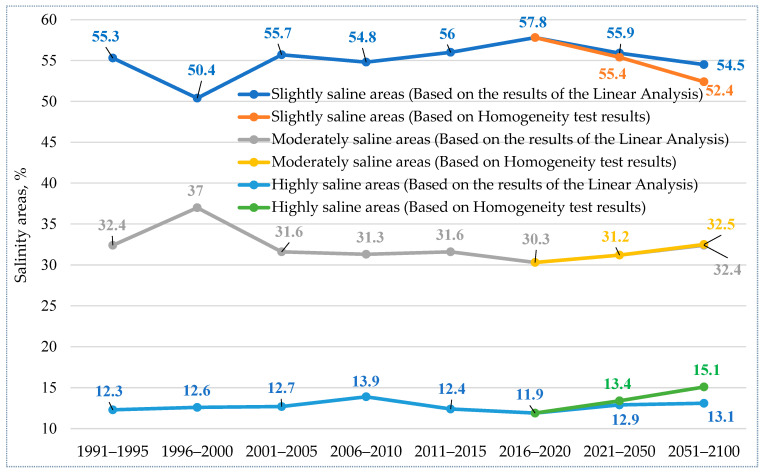
Projection of salinity areas.

**Table 1 ijerph-19-08794-t001:** Physical and chemical properties of the soils in the Khorezm region, Uzbekistan.

Depth (cm)	Texture (%)			Bulk Density (g cm^−3^)	Soil pH	Soil Organic Carbon (%)	Total N (%)	Available Phosphorus (mg kg^−1^)	Exchangeable Potassium (mg kg^−1^)
	Sand	Silt	Clay						
0–10	24	59	21	1.25	6.21	0.44	0.06	29.2	98.5
10–20	34	52	19	1.34	6.18	0.36	0.06	27.1	95.0
20–30	27	64	14	1.39	6.19	0.30	0.05	23.8	89.3
30–60	31	66	10	1.48	6.95	0.27	0.04	21.3	81.4
60–90	52	46	9	1.54	6.26	0.23	0.04	19.7	76.8

**Table 2 ijerph-19-08794-t002:** Salinity areas in the Khorezm region from 1991 to 2020.

Salinity Levels (%)			Year			
	1991–1995	1996–2000	2001–2005	2006–2010	2011–2015	2016–2020
Slightly	55.3	50.4	55.7	54.8	56	57.8
Moderately	32.4	37	31.6	31.3	31.6	30.3
Highly	12.3	12.6	12.7	13.9	12.4	11.9

## Data Availability

Not applicable.

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
