# Peer review of "Assessment of Soil Salinity Changes under the Climate Change in the Khorezm Region, Uzbekistan"

_ijerph, 2022, doi:10.3390/ijerph19148794_

Round 1

Reviewer 1 Report

This manuscript presented a prediction of soil salinity under climate change in the Khorezm Region of Uzbekistan. Some most important factors linking climate change and shifts of soil salinity were considered in the methodologies used. Soil salinity is by all means a very important property of soil and has huge implications for agricultural productivity and well-being of local people. This study projecting soil salinity changes in a changing climate provide useful information for not only researchers, but also policy makers. Therefore, the reviewer believes this manuscript deserves publication in this journal after some revisions to address a few deficiencies, as detailed below:

1) This study predicted the percentage changes of soils with different saline levels, whereas the implications were not discussed. Soils with different saline levels would have different impacts in agricultural production, so it would be better to talk about these effects, and the implications of soil salinity changes at different levels.

P    2) Please correct the first cell in Table 3.

      3) As an important source of irrigation water, glacier melting is and will keep changing drastically under climate change. Glacier melting will affect irrigation water and its qualities, and subsequently affect soil salinity. Therefore, it would be necessary to include some discussions, for which the literatures below should be cited:

Abbas et al., 2020. Water Supply. Physicochemical properties of irrigation water in western Himalayas, Pakistan.

Sun et al., 2017. Environmental Pollution. The role of melting alpine glaciers in mercury export and transport: An intensive sampling campaign in the Quagaqie Basin inland Tibetan Plateau.

White et al., 2014. Water Resource Management. The Impact of Climate Change on the Water Resources of the Amu Darya Basin in Central Asia.

Author Response

The authors thank the reviewer for time and the effort spent to improve the manuscript. The efforts are greatly acknowledged. We agree in general by all points raised by the reviewer and have revised the manuscript accordingly. Please see attached the document for our detailed responses.

Reviewer 2 Report

The manuscript presented a comprehensive statistical framework for predicting soil salinity dynamics of the Khorezm Region of Uzbekistan using the Homogeneity test and linear regression model. The topic is worth to be investigated and is of potential interest to readers of International Journal of Environmental Research and Public Health. However, I have some concerns that need to be addressed before consideration for publication.

This manuscript used the Homogeneity test and linear regression model to evaluate the change in soil salinization, so the research progress of relevant models should be introduced in more depth in the introduction.

Figure 1. The map of the Khorezm region, Uzbekistan is too blurry and needs a clear image file

Table 1. Physical and chemical properties of the soils in the Khorezm region, Uzbekistan

Are these samples duplicated? Deviations should be provided.

The content in Table 2 is very small, it is suggested to delete it, the content can be reflected in the article

Table 3. Salinity areas in the Khorezm region, Uzbekistan

The oblique lines in the Table need to be re-edited.

Line 201-216: Some symbols in the formula are not explained, please add them.

A comprehensive analysis of the climatic trends in the study sites was assessed in the first step. In this context, we analyzed the mean annual air temperature at Khiva and Urgench meteorological weather stations based on the Homogeneity test.

This is the basis for determining the points (years) of average temperature change (1984 in Khiva and 1976 in Urgench), but I think the determination of these points is very important for this article, so more explanation needs to be added.

I think Figure 2 and Figure 3 can be combined into one diagram.  (Figures 5, 6, and 7 can also be combined)

Table 1 showed the Physical and chemical properties of the soils in the Khorezm region, Uzbekistan

Do these soil physical and chemical properties affect changes in soil salinity? It doesn't seem to be well discussed in this article and should be added to improve the quality of the article.

Author Response

(The authors gave the same response as above.)

Reviewer 3 Report

This work is very interesting and concerns salinity changes as a result of climate change in one of the regions of Uzbekistan. However, despite its regional scope, its overtones concern global problems with similar climatic zones. Statistical tests are the strong point of the work. Implementing multi-variate statistical approaches to estimate and evaluate soil salinity can be useful for better decision making in the management of the agricultural regions.

Detailed comments:

• Figure 3 shows the change in the temperature trend since 1976. The figure lacks the statistical parameters (p value) how significant they are. There is generally no comment on such a strong change in the trend line (since 1976) beyond the well-known fact of global warming

• The list of literature requires the unification of the names of journals - full names or accepted abbreviations; eg line 313 - European Journal of Soil Science (full name) and line 318 - Irrig Drain (abbreviation of the name - Irrigation drainige). Throughout the list, the authors use alternately full names of journals or abbreviations of these names

• The selection of references is quite modest - only 30 items. In order to increase the value of the work, the selection of references should be wider, the more so as it concerns the broad problem of soil salinity

• How representative are the given soil properties in table 1 - they represent a very large region - are these average values, the number of analyzes, etc.

Author Response

(The authors gave the same response as above.)

Round 2

Reviewer 2 Report

The author has made corrections according to the revised comments and corresponding problems. I personally think that the manuscript has met the requirements for publication. It is recommended to accept.

Reviewer 3 Report

The authors referred to all my comments presented in the review. I believe that the manuscript has been sufficiently improved and fully recommend it for publication in the International Journal of Environmental Research and Public Health.

This manuscript is a resubmission of an earlier submission. The following is a list of the peer review reports and author responses from that submission.

Round 1

Reviewer 1 Report

The manuscript "PROJECTIONS OF SOIL SALINITY CHANGES UNDER THE CLIMATE CHANGE IN THE KHOREZM REGION". ID: sustainability-1716804

Dear Authors,

I have written my comments for your manuscript on the pdf file pages. So, please download it and revise the manuscript according to the comments. Good luck.

Regards.

Reviewer 2 Report

Soil salinity is a process that has clear causative factors associated with it. Annual Mean air temperature is not one of them. Trying to relate two variables that are not physically related is bad science. The whole premise of the paper is questionable. Any correlation between these variables must be spurious and meaningless. If soil salinity could be related to air temperature then why not with relative humidity, airspeed, population, and many others.

Reviewer 3 Report

            The manuscript entitled “PROJECTIONS OF SOIL SALINITY CHANGES UNDER THE CLIMATE CHANGE IN THE KHOREZM REGION” (reference sustainability-1716804) projected the future soil salinity changes under climate change using some statistical methods. The topic is interesting. The main concerns to this study are:

  1. Many projected future climate data under different climate change scenarios are available now, why this study didn’t use these resources to project the future soil salinity change?
  2. How reliable are the projected soil salinity changes in this study? The uncertainties and/or the validation of this work should be discussed.